# Method of Choice: A Fluorescent Penetrant Taking into Account Sustainability Criteria

**Andrzej Pacana** [1],*, **Dominika Siwiec** [1] **and Lucia Bednárová** [2]

1    Faculty of Mechanical Engineering and Aeronautics, Rzeszow University of Technology,
     al. Powstancow Warszawy 8, 35-959 Rzeszow, Poland; d.siwiec@prz.edu.pl
2    Faculty of Mining, Ecology, Process Control and Geotechnologies, Technical Unviersity of Košice,
     042 00 Košice, Slovakia; lucia.bednarova@tuke.sk
*    Correspondence: app@prz.edu.pl; Tel.: +48-17-8651-390

**Abstract:** To conduct, in an effective way, the non-destructive testing (NDT) of products—in particular, the fluorescent penetrant inspection (FPI)—remains a challenge. Therefore, the aim of this work is to propose the method of support in the choice of a fluorescent penetrant to be used in FPI research. In the results of the usage of the proposed procedure, it is demonstrated that it is possible to reduce the negative impacts on the environment by FPI processes (through sustainability), while including other criteria, i.e., financial, security, productive (Industry 4.0), and societal (Society 5.0) criteria. The essence of the proposed method is to integrate two methods of decision support. These were the analytic hierarchy process (AHP) method and the cost–quality analysis (AKJ). Using the AHP method, the quality level of fluorescent penetrant (to the satisfaction of the customer)—which included the sustainability criteria—are calculated. These criteria include natural environment, reactivity, combustibility, level of sensitivity, and type of washing (emulsification). Then, with the help of the AKJ, the most favorable penetrant—in terms of quality and cost—is calculated and, thus, indicated. This choice must include the concept of sustainable development. Therefore, this method can be used to choose fluorescent penetrants in manufacturing and service enterprises which carry out FPI.

**Keywords:** sustainability; Society 5.0; Industry 4.0; production engineering; fluorescent research

## 1. Introduction

Product control of quality and properties is possible by destructive testing (DT) and non-destructive testing (NDT) [1,2], but non-destructive testing is more frequently used, less costly, and has a much larger range of applications than destructive testing [1,3]. This results from the possibilities of NDT research, consisting of product control of quality and properties without destroying the product in question [1,4,5]. Among many NDT techniques, there is the occurrence of, among others, visual inspection (VI) [6]. Next, for example, is the ultrasound inspection, in which discontinuities are identified using acoustic waves, which have their main applications in, e.g., the control of the composite product, joints, rails, and to control on an assembly line where the same product must be controlled multiply [6,7]. Other research suggests thermographic and infrared thermographic tests that are performed via emitted heat radiation, and this research has its main applications in the control of narrow width products and results from the lack of a possibility to identify possible deeply embedded defects [6,8]. Yet another kind of NDT research is, for example, radiographic research (RT), in which, using radiation, and image is obtained, which is then X-rayed using radiographic film in order to obtain a radiogram; however, in this case, the results from research are highly dependent on the interpretation of the person who conducted research [9]. In NDT controls, the most commonly used technique—used in over 90% of metal product testing, including, for example, in the aviation industry [10]—is fluorescent penetrant inspection (FPI). In the

case of the mentioned aviation industry products—the testing of which involves highly demanding processes, in terms of quality and safety [3,10]—these control tests are conducted at least once during their production [10], independently from the visual control used [11]. During the fluorescent penetrant inspection, fluorescent penetrants are used, and examples of consumption in FPI control, in one of Polish enterprise, averaged from 1000 to 1400 dm$^3$ per year, in case of products fully immersed in penetrants. Despite a significant share of FPI research—mainly in product controls [10,12]—carrying fluorescent penetrants effectively and according to sustainability standards, challenges nevertheless remain. The main limitations in carrying out consistent, reliable, and efficient FPI research result from the negative impact of fluorescent penetrants on the natural environment [13,14], hazards resulting from their reactivity and flammability [15–19], and also from human factors which may occur in the processing, control, and assessment of products [10,20,21], which—in the light of other NDT studies—was considered important criteria of analysis. It was considered that it is possible to introduce sustainability criteria into the method that allows decision-making about the selection of penetrants for FPIs, and such decisions will be part of the concept of Society 5.0 and Industry 4.0. Therefore, the aim of this work is to propose the method of support in the choice of a fluorescent penetrant to be used in FPI research. It is possible to implement the proposed method in the framework of digital network systems, presenting the opportunity to develop enterprises towards Industry 4.0 [15].

These negative environmental aspects of penetrants result mainly from the presence in these of petroleum; after getting into waters, petroleum causes long-term negative changes in the aquatic environment [22]. In order to reduce the negative impact of fluorescent penetrants on the natural environment, the methods of the authors of work [13] were applied as fluorescent penetrant in the suspensions based on *E. coli* bacteria (*Escherichia coli*), which is a friendly substance to the environment. In work [14], using a fluorescent penetrant which detects cracks in welds, the semiconductor quantum dots (QD) were applied, which are nanometric structures fluorescing under UV light. Application of the suspension *E. coli* bacteria [13], or semiconductor quantum dots (QD) [14], instead of a traditional fluorescent penetrant, enabled the authors to limit the negative impacts of the fluorescent penetrant on the environment, and also reduce, for example, the number of steps required during the preparation and cleaning of products and the costs of penetration tests [13,14]. Additionally, in order to reduce the negative impacts of the FPI on the natural environment and increase the reliability and performance of FPI research, in the publications [10,20,21], the effectiveness of an automatic fluorescent penetrant inspection was analyzed. Similarly, in work [20], to increase the reliability and effectiveness of the FPI research, the automatic control of products was proposed by applying the machine learning algorithm named "Random Forest" (RF). This serves to detect defects, which were first identified by fluorescent penetrant inspection. It was demonstrated that this technique, among others, with a small number of repetitions in the learning process, allowed the authors to distinguish correctly the product incompatibility with an accuracy comparable to that of human work [20]. Additionally, in publication [10], the method of the modification of this technique was analyzed based on individual decision trees, and it has been shown that it is possible to improve the learning process of defects, while obtaining results similar to those obtained from RF machine learning. In publication [21], the automatic control system (AAIS) was proposed, using image processing and pattern recognition methods, allowing automatic identification, measurement and classification of defects based on pictures from the fluorescent penetration inspection. The actions taken to automate FPI research, heading towards the concept of Industry 4.0, reduced the consumption of materials and penetrants, in comparison to manual FPI; this, in turn, was a part of the idea of aiming towards sustainable development [23–26].

The literature review showed that previous activities have aimed to achieve reliable and effective fluorescent penetrant inspection [5,20,21]. Furthermore, it has been attempted to reduce the negative influences of the fluorescent penetrants on the natural environment [13,14,22]. However, a solution has been not achieved which would allow one to unambiguously choose the FPI method which most effectively takes into account reliability, efficiency, and the environment. Production engineering

deals with, among others issues, solutions to such problems. However, it has been observed that previous works also consider the idea of sustainability and the concepts of Society 5.0 and Industry 4.0, where sustainable development means improving the living conditions of society, while maintaining the ecological integrity of the planet for future generations. Sustainable development plans are intended to ensure public wellbeing while respecting ecological integrity [27]. The above-mentioned concept Industry 4.0 promotes the integration of the smart machines, systems, and making changes in production processes, having the aim to increase production efficiency and the possibility of flexible changes of assortment. In general terms, it concerns the production process [28] and the concept of a so-called smart factory, which use, among others, cyber-physical systems (CSP), the internet of things (IoT), and cloud computing. Using the Industry 4.0 concept, enterprises having the possibility to achieve higher efficiency and autonomic possibilities adapting to needs [10,29], which—in the character of search-reliable FPI research—has related to activities of automatic production, whereas the Industry 4.0 concept is coexistent with the Society 5.0 concept, which was developed in 2015 and is also known as the imagination society concept [30]. The aim of Society 5.0 is to create a sustainable society in which people can live safely and comfortably [24], to balance economic progress with the search for solutions to social and environmental problems [23,30]. The search for methods of achieving reliable, effective, and relatively harmless (for the environment) of FPI research, is crucial in the achievement of the concepts of both Industry 4.0 and Society 5.0. It concerns the search for effective, concentrated, and divergent solutions of problems which involve both society and the engineering complexity. This also concerns the application of changes in the environment and in the technological processes, which force the need to search for modern methods supporting the achievement of reliable FPI research.

In the case of the changes to society and industry caused, despite the previous research as a part of improving FPI, there is still a need to search for a method for choosing a penetrant in FPI, and therefore this remains a current research problem. Due to the fact that significant quantities of penetrants consumed are industrially used, often by automatic FPI research [1,3,10], there remains the need to develop and apply a method which supports the choice of fluorescent penetrants in FPI. Therefore, as a part of the development of enterprises towards of Industry 4.0., it is important to enable the making of the choices of penetrants as a part of digital networks, and also for developing a program which the decision-making, for example, Expert Choice [15,31]. Therefore, contemporary methods, in addition to including reliability, efficiency, and sustainable development, should, at the same time, bring the current research closer to the concepts of Industry 4.0 and Society 5.0.

The aim of this work is to propose the method of support in the choice of a fluorescent penetrant to be used in FPI research. As a result of the usage of the proposed procedure, it will be possible to reduce the negative impacts of FPI processes on the environment in (through sustainability) while including other criteria, i.e., financial, security, productive (Industry 4.0) and societal (Society 5.0) criteria. In this context, the proposed method refers to Society 5.0, whose purpose is to facilitate public wellbeing while respecting ecological integrity [32]. An important benefit of the proposed method (which is a combination of analytic hierarchy process (AHP) and cost–quality analysis (AKJ)) is the possibility of applying it within digital network systems, providing an opportunity for the development of enterprises towards the concept of Industry 4.0 [33]. The method was proposed in enterprise located in in Podkarpacie, Poland, in which fluorescent penetration inspection is conducted for products which the aviation industry manufacture. Choosing the enterprise to research was conditioned by the significant consumption of fluorescent penetrants during FPI (an average consumption of two fluorescent penetrants was around 1000 $dm^3$ to 1400 $dm^3$/year) and the motivation of the enterprise to achieve reliable and efficient FPI research, and also FPI which is compatible with the principles of sustainable development, which will introduce the enterprise to the concepts of Industry 4.0 and Society 5.0.

## 2. Materials

The fluorescent penetrants, used relatively often in industrial practice, were selected in order to test the proposed method and were characterized by the average level of sensitivity (two sensitivity levels) [34]. These penetrants were: ZL-2C and ZL-60D (Magnaflux), FP-922 (Met-L-Chek), Ardrox 970P23 (Chemetall), FP-22B, and MH-3A (Sherwin Incorporated). According to the needs of the research, these were marked randomly, in a conventional manner, from P1 to P6. The choice of the type and number of penetrants was conducted by experts (the manager of NDT in the enterprise which was selected to test the method), as well as by the authors of this work, who were using the brainstorm and multiple voting technique, resulting from their knowledge and experience, as well as the average level sensitivity (in the ranges: very low $\frac{1}{2}$; low 1; average 2; high 3; very high 4) [34] required by the enterprise for detecting the defects of the product, and from the criteria and specifics of the control. The penetrants selected for the analysis are typically used for testing castings, forgings, rough surfaces, nickel, steel, or titanium products. Defects detected by these penetrants include, for example, cracks, delamination, porosity, seams, metal leaks, composites, synthetic materials, and some plastics. Information about the analyzed penetrants includes their publicly available safety data sheets.

## 3. Methods

The proposed method of choosing a fluorescent penetrant uses two integrated methods: the AHP (analytic hierarchy process) and AKJ (cost–quality analysis), which are supported by addition techniques, i.e., the brainstorming (BM) and multiple voting techniques. The integration of the methods of the AHP and AKJ resulted from their effectiveness when integrated with other techniques [35–37], in order make the best choices and calculate levels of product quality [38], including the cost aspect [36,38].

The choice of the AHP method, which is one of the methods used to solve the decision problem, was influenced by the fact that the criterion of a choice of penetrants can be either quantitative or qualitative, and can also be analyzed either subjectively or objectively [39–41], which result from modeling in the contexts of uncertainty and risk [37,42]. Furthermore, the choice of penetrant, resulting from the effectiveness of the AHP method in decision-making based on both subjective and objective assessments, can be made by an expert group [43]. Furthermore, from the possible implementation of this method is as a part of digital network systems, as well as through the use of, for example, Expert Choice software, which is a graphical, intuitive interface used for the assessment and determination of priorities, in addition to controlling the sensitivity of assessments [31], thus providing the opportunity to develop enterprise towards the concept of Industry 4.0 [15]. In the AHP method, the pro-environment criteria is introduced, which represent the idea of sustainable development and the concepts of Society 5.0 and, partly, Industry 4.0. In this method, the quality level of fluorescent penetrants is assessed; however, this does not include the cost of penetrant purchasing. In view of this, the AHP method was integrated with the AKJ method: an uncomplicated technique used to determine the best choice in terms of cost–quality [36,44,45]. The use of AKJ results from the possibility it provides of linking the costs of penetrant purchasing with the quality of such penetrants. Through this, it becomes possible to choose a fluorescent penetrant in terms of its cost–quality.

The proposed method of choosing a fluorescent penetrant was separated into six stages, which are represented in next part of the work.

### 3.1. Specifying the Purpose

The first stage of the proposed method is the choice of the aim of the research. In this case, the aim of the work was to support the decision process through the choice of a fluorescent penetrant for FPI research, while including sustainability criteria. As a part of the aim, it becomes possible to reduce the negative impacts of the FPI processes on the environment (through sustainability), while including

other criteria, i.e., financial, security, productive (Industry 4.0) and societal (Society 5.0) criteria. Therefore, the choice of the aim takes into account the reliability and effectiveness (corresponding to cost) of the research and criteria in the context of the concepts of Society 5.0 and Industry 4.0.

### 3.2. Specifying the Set of Criteria

The second stage of the method is determining the set of criteria to be used, on which the choice of the fluorescent penetrant will be based. Based on the above-mentioned aim, the criteria, in context of the idea of sustainable development, were chosen [10,13,14,20]. Additional criteria adopted—which are included in the proposed method (except for standard ones, i.e., the level of penetrant and penetration time)—are its influence on the natural environment, reactivity, and combustibility. Based on the adopted goal, the choice of criteria was made in the context of the ideas and goals of sustainable development, i.e., [27]:

> Goal 12: to ensure sustainable consumption and production patterns;
> Goal 13: take urgent action to combat climate change and its impacts;
> Goal 14: conserve and sustainably use the oceans, seas, and marine resources for sustainable development.

The selected criteria are the main criteria of the National Fire Protection Association (NFPA) and the Hazardous Materials Identification System (HMIS) system [18,22,46,47], which are included in the analysis of chemical hazards, in the context of their sustainability development [46]. These systems are a numerical assessment of threats, where 0 = low threat and 4 = high threat. They are used to label hazardous substances and mixtures, which are also included in the data sheets of, e.g., fluorescent penetrants [18,22]. On the other hand, in the context of the cited goal 12, it was considered important to take into account, for example, the environmental and social aspects of the choice of environmentally friendly products during production.

The first criterion is the influence on the natural environment (so the degree of impact the penetrant on the natural environment) [10,13,14,20], was analyzed and efforts were made to limit its negative impacts, for example by replacing it with E-cola bacteria [13] or semiconductor quantum dots (QD) [14]. However, fluorescent penetrants are still used, which are not replaced by less environmentally harmful substitutes, by which the method of choice of these penetrants for the FPI was not pointed, which would include their influence on the natural environment. Therefore, this criterion was recognized for reasonably taking into account the proposed method—especially as penetrants, after getting into water, have negative, irreversible effects on the natural environment [22]—which is a part of sustainability goals, i.e., 13 and 14.

The second criterion is the reactivity of fluorescent penetrants, which is the ability to react with other chemical compounds under certain conditions, which manifests itself especially in highly fluorescent penetrants [16]. Penetrants evaporate and reactivate with liquid oxygen (LOx), and other strongly oxidizing material [48], creating the explosive reaction [15,16,49]. The before-mentioned explosive reactions are created as detonations resulting from shock or vibration [17]. Additionally, penetrants which contain fluoride may be reactive with aluminum or aluminum alloys during machining to remove cracked areas. Therefore, an important part of effective FPI research is the use of such a fluorescent penetrant that is acceptable for use, among others, on aluminum surfaces moistened with liquid oxygen. [17].

The literature review shows that the reactivity of penetrants with liquid oxygen was reduced, for example, by using as a penetrant the water or water with glycol [17]; however, these substances, after evaporation, still have reactivity with LOx. Then, the trifluorochlorethylene polymer was used [16], but in work [49], as a cheaper substance than trifluorochlerethylene polymer, the use of hexachlorobutadiene-1,3 was proposed. In turn, the authors of work [17] used, for example, non-volatile fluorocarbon liquid and volatile thinner, which included dissolved indicator dye. Despite this, reactive penetrants are still on the market [15], and the method which includes this criterion in choosing

the penetrants to FPI research was not indicated. Additionally, criterion reactivity and the next criterion, i.e., combustibility, are one of the basic criteria included in analyses of achieving sustainable chemicals, and therein, the criteria of NFPA and HMIS systems [18,46,47], were considered for importance in proposed method as part of sustainability (for example, goal 12) [27].

The before-mentioned third criterion is the combustibility of fluorescent penetrants, the ability of penetrants to sustain fire, wherein the fluorescent penetrant, being a flammable liquid, has a flash point not exceeding 93 °C [18]. Efforts were made to limit the flammability of penetrants by using non-flammability penetrants [47]; despite this, it is a fact that most chemicals used in NDT research are still flammable [19], which generates the needs to include this criterion in the choice of penetrants as part of sustainability (for example goal 12) [27].

It was concluded, therefore, that these criteria, i.e., influence on natural environment, reactivity and combustibility, are important main criteria which should be included in the proposed method in the context of sustainability [15–19,22,27,47,48]. However, due to the case of constant changes in the environment and technological processes occurring, it was considered that it will be effective within the proposed method to not limit the number and type of criteria. Both the environmental and other criteria should be chosen depending on the needs of the entity, using the method.

Selection of criteria, in accordance with the adopted method of teamwork should be done by a team of experts with 4 to 10 employees. They should be employees from, for example, the control department, or also from the purchasing department, which, according to the head, has the competencies and qualifications to choose the penetrant. To choose the assessment criteria in the AHP method, the team should apply the technique that they are allowed to generate. A commonly known technique which can be used in this cause is brainstorm (BM) [49]. The brainstorming technique should be started from formulating the problem and establishing the team (the before-mentioned expert team). Then, as part of the methodically implemented technique [49], the list of criteria should be created. In the next part of the brainstorming, the ranking list of the reported criteria should be created. In this aim, the expert team can use, for example, the commonly known and uncomplicated technique of multiple voting. Another method that can be used in this brainstorming step is the Suzuki method. After creating the ranking list, the team determines the criteria that will be taken into account when assessing penetrants. Please follow the rule that, as parts of these criteria, aspects related to Society 5.0 and Industry 4.0 must be considered. The number of criteria, in the collection of criteria, should be no more than 9 [42]. This follows from the recommendations of the AHP method (used in the proposed method), where, according to the Miller psychological framework, the simultaneous pairwise comparison 7 ± 2 criterion is recommended [42,49].

### 3.3. Selecting the Penetrants to Analyze

The third stage of the method is the choice of the fluorescent penetrants to analyze. The selection of penetrants is made by a team of experts: the same ones who, in the second stage, chose the criteria for their assessment. In terms of the various characters of FPI research and varieties of fluorescent penetrants available on the market, the number and type of fluorescent penetrants in the proposed method can be any, depending on the entity using the method. It would be advisable for the team, based on knowledge and experience, to choose penetrants that differ from each other in the context of research conducted in the enterprise.

### 3.4. Determination of the Quality Level of Penetrants

The fourth stage of the method is the appointment of the quality level of fluorescent penetrant. The appointed quality level of fluorescent penetrant relies on performing calculations with the AHP method. The appointment of a quality level of penetrants was divided into three essential sub-stages (Sections 3.4.1–3.4.3).

During the determination process of the quality level of the fluorescent penetrant, the proposed (by Saaty, creator of the AHP method) scale was applied, i.e., from 1 to 9, where 1 means equal importance, and 9 absolutely more important [39–41].

### 3.4.1. Determination of Criteria Weights

The first step is to assign ratings to all the criteria included in the set of criteria, according to the Saaty scale. The criteria are assessed by the entity using the method: the team of experts who selected these criteria (Section 3.2) and subsequently the selection of penetrants (Section 3.3). The assessments of criteria are made according to the multiple voting technique, so that is, by each of team member on the suggested by the Saaty scale (i.e., 19) or some other, for example, 1–5 or 1–7 [49] casts a vote on individual criteria from a set of criteria, judging by how many times one criterion is more important than the other. During the multiple voting, in view of the different criteria and also different specificities of enterprises, it is recommended to prepare, for the team of experts, the tables on the pattern of these tables used in FMEA, facilitating and, at the same time, standardizing the assessment of criteria. It is possible, using a special questionnaire, to capture all the grades of decision-makers, for example, as in work [31]. After making an assessment of all criteria, it is necessary to make a list of criteria ordered by importance, where the resulting assessment is the median of the results of the votes for each criterion for each penetrant.

The second step is a pair comparison of the ratings assigned to the criteria in the first steps and saving them in the Saaty matrix, i.e., $S = (S_{ij})$, where: $i, j = 1, 2, \ldots, k$; with proportion of weights $i$-th and $j$-th criteria (1), being a square matrix of the order of n × n, where n = the number of criteria (2), and is therefore called the dominance matrix [40]:

$$S_{ij} \approx \frac{w_i}{w_j}, \text{ where } i, \ j = 1, \ 2, \ \ldots, \ k. \tag{1}$$

$$S_{ij} = \frac{1}{S_{ij}}, \text{ where } i, \ j = 1, \ 2, \ \ldots, \ k. \tag{2}$$

In matrix *S*, always on the diagonal is grade 1, meaning that the criteria compared are equivalent to each other; therefore, above the diagonal are the results of the comparison of the two criteria, and below the diagonal, the inverse comparison values [40].

The third step is to calculate the criteria weights based on the Saaty matrix. The calculation of the criteria weights is done by the geometric mean of the rows of matrix *S*, which are subsequently normalized (3) [40]:

$$w_i = \frac{\left[\prod_{j=1}^{k} S_{ij}\right]^{\frac{1}{k}}}{\sum_{i=1}^{k}\left[\prod_{j=1}^{k} S_{ij}\right]^{\frac{1}{k}}}, \ for \ i \ldots 1, \ldots. \ k. \tag{3}$$

The correctness of the calculations is proven by the sum of the weights of all the values for a given criterion level, which should be equal to 1.

The fourth step is to check if the received values did not violate the principle of the constancy of preferences [50]. This consists of calculating the inconsistency factor ($\lambda_{max}$) (4), comparison matrix compliance factor *CI* (5) and compliance relationship *CR* (6) [40,41]:

$$\lambda_{max} = \frac{1}{w_i} \sum_{j=1}^{k} w_{ij} w_j \tag{4}$$

$$CI = \frac{\lambda_{max} - n}{r(n-1)} \tag{5}$$

$$CR = \frac{CI}{r} \tag{6}$$

where: $\lambda_{max}$ = inconsistency factor; $n$ = number of criteria (in Section 3.4.1) or penetrants (in Section 3.4.2); $r$ = average random index value for n according to Saaty [50].

Full compatibility of results are when $\lambda_{max} = n$, $CI = 0$ and $CR = 0$. However, compatibility of results is acceptable for $\lambda_{max}$ near $n$, for $CI < 0.1$ and $CR < 0.1$ [40]. Failure to achieve full or acceptable compliance of the results obtained indicates that the marks were awarded in an inconsistent manner. In this case, the process should be repeated from step one and repeated until the obtained results are fully compatible or acceptable. After achieving full or acceptable compliance of results, a further determination of the quality level of penetrants should be carried out.

### 3.4.2. Determination of Penetrants' Weights in Terms of Criteria

An assessment and weighting should be carried out for each selected fluorescent penetrant in terms of all criteria included in set of criteria. At the beginning, the assessments should be assigned to all the criteria included in the developed set of criteria (second stage). This process is carried out for all analyzed penetrants. Assessment of criteria according to the accepted Saaty scale is made by an entity using the proposed method. The procedure is analogical to the procedure present in Section 3.4.1.

### 3.4.3. Determination of the Value of the Quality Level of Penetrants Q

The first step is making criteria rankings based on the obtained results of criteria, ordering criteria from the most important to least important, where a maximum weight of criterion means that criterion is the most important (position 1 in ranking).

The second step is a multiplication of the values of weights matching the given criterion by all values of penetrants weights in terms of given criterion weight.

The third step is to sum the values of all criteria in terms of given penetrant, which were obtained in the previous step. An obtained sum of values for each penetrant, which are real numbers, are equal with a quality level of penetrants (Q). It is possible to compare the Q values, which allows an indication of the best fluorescent penetrant in qualitative terms. The most beneficial penetrant in qualitative terms is the penetrant about the maximum value of quality level Q.

Seemingly difficult to calculate, the AHP method can be performed using computer-aided techniques, for example Matlab or program R [51,52]. Details of realizing the AHP method can be found in the literature (for example, [39–41]).

### 3.5. Determination of the Cost–Quality Level of Penetrant

The fifth stage of the proposed method is determining the level of cost–quality the fluorescent penetrants by cost–quality analysis (AKJ). This stage was spread over three steps (Sections 3.5.1–3.5.3)

### 3.5.1. Conversion of Q Value into Percentage Value

In the first step, it is necessary to convert the obtained (in AHP) method into the quality level of penetrants (Section 3.4.3–third step), which are real numbers on the percentage values. In this aim, the obtained values of the quality level of penetrants (Q) should be multiplied by a value of 100.

### 3.5.2. Determination of the Cost of Penetrants Purchasing

In the second step, it is necessary to determine the cost of purchasing a required number of packages of the penetrants, taking into account the penetrant packaging capacity.

### 3.5.3. Calculation the AKJ Indicators

In the third step, the indicators of cost–quality analysis were calculated for each of penetrant analyzed (7–20).

First is quality cost $c_k$ (7), which is a quotient of the cost of the purchase of the required number of packages of penetrant (i.e., Section 3.5.2) and the level of quality, $Q$: this penetrant expressed as a percentage (i.e., Section 3.5.1) [35–37,44,53]:

$$c_k = \frac{K}{Q} \tag{7}$$

where: $K$= cost [€]; $Q$ = level of quality [%].

Second is the indication relativized cost $k$ (8), which is a quotient of two differences: the difference between the maximum cost of the total costs of analyzed penetrants ($K_a$) and the cost of purchasing t required the number of packages of penetrant ($K$), and difference between the maximum cost of the total costs of the analyzed penetrants ($K_a$) and the minimum cost of the total cost of the analyzed penetrants ($K_i$) [35–37,44,53]:

$$k = \frac{K_a - K}{K_a - K_i} \tag{8}$$

where: $K_a$ = maximum cost for AKJ [€]; $K_i$—minimum cost for AKJ [€]; $K$—cost of penetrant [€].

Third is the indication cost-quality proportionality $E$ (9), which is a quotient of relativized cost $k$ (8) and quality level $Q$ expressed as a decimal fraction $q$ (10) [35–37,44,53]:

$$E = \frac{k}{q} \tag{9}$$

$$q = \frac{Q}{100} \tag{10}$$

where: $k$ = relativized cost k, as in the Formula (8) [€]; $q$ = quality level Q expressed as a decimal fraction.

Fourth is the indication of relativized cost $c$ (10), which was calculated according to [35–37,44,53], so the quotient of the two differences. The first difference is the maximum quality indicator $c_{ka}$ from all quality indicators $c_k$ calculated according to Formula (7) minus quality indicator $c_k$ for a given penetrant. The second difference is the maximum quality cost indicator $c_{ka}$ from all quality cost indicators $c_k$ calculated according to Formula (7) minus minimum quality cost indicator $c_{ki}$ from all quality cost indicators $c_k$ calculated according to Formula (7). The quotient these differences is determined by the Formula (11):

$$c = \frac{c_{ka} - c_k}{c_{ka} - c_{ki}} \tag{11}$$

where: $c_{ka}$ = maximum quality cost indicator in AKJ; $c_{ki}$ = minimum quality cost indicator in AKJ.

The fifth is the indication of decision function indicators $d_0$ (12) or $d_1$ (13), which are comparable with the universal, single scale of relative states, giving the possibilities of making a preliminary decision about the choose one of comparable a few variants [35–37,44,53]:

$$d_0 = 0.5 \times E, \quad for\ E = 0 \div 1 \tag{12}$$

$$d_1 = 0.5 + 0.5 \times \left(1 - \frac{1}{E}\right), \quad for\ E > 1 \tag{13}$$

where: $E$ = indication cost-quality proportionality (Formula (9)).

As stated by Kolman, author the work [37] (creator of the AKJ method), using these variants of decision function $d_0$ and $d_1$ is possible by the condition is met (14):

$$Q \geq Q_{gr} \tag{14}$$

where: $Q_{gr}$ = the minimum acceptable value of quality level ($Q$).

For these coefficients, the indication of settlement for technical preference $R_t$, which is calculated for the authors of works [35–37,44,53], from dependence (15) is:

$$R_t = \frac{\alpha q + \beta d + \gamma c + \delta k}{\alpha + \beta + \gamma + \delta} \tag{15}$$

where: $q$ and $k$ are as in Formula (9); $c$ is as in Formula (11); $d$ is as in Formulas (12) or (13).

In turn $\alpha$, $\beta$, $\gamma$ and $\delta$ are the coefficients of importance which, according to Kolman (author of AKJ method), can be taken in proportion (16) [37]:

$$\alpha : \beta : \gamma : \delta = 8 : 4 : 2 : 1 \tag{16}$$

For these coefficients, the settlement for technical preference $R_t$ takes the form (17) [35–37,44,53]:

$$R_e = 0.0667(8k + 4c + 2d + q) \tag{17}$$

Seventh is settlement indicator for economic preferences, $R_e$ (12), which was resulted from dependence (18) [35,37]:

$$R_t = \frac{\alpha k + \beta c + \gamma d + \delta q}{\alpha + \beta + \gamma + \delta} \tag{18}$$

where: markings as in the Formula (15).

After taking the proportions as in the Formula (16), the settlement indicator for economic preference $R_e$ is obtained (19) [35–37,44,53] by:

$$R_t = 0.0667(8q + 4d + 2c + k) \tag{19}$$

The indicators $R_t$ and $R_e$ have application to the verbal interpretation the result of AKJ, which follows from the universal, single scale of relative states. Therefore, in aim of making a decision that is an indirect choice, it is necessary to calculate the averaged decision-making indicator $R_d$ (20) [35–37,44,53]:

$$R_d = \frac{R_t + R_e}{2} \tag{20}$$

The indicator $R_d$ also has applications to the verbal interpretation of AKJ result and making decisions according to the universal, single scale of relative states. Obtained values of average decision-making indicator ($R_d$) are the value levels of cost-quality of analyzed products. There is a rule that the higher $R_d$ value, the cost-quality of penetrants is better.

*3.6. Choosing the Most Favorable Fluorescent Penetrant*

The last stage, the sixth stage of the proposed method, is the choice of fluorescent penetrant which includes sustainability criteria. The fluorescent penetrant with the highest value of the decision-making indicator ($R_d$) is the most favorable. This choice results from the idea of the proposed method, in which the best choice (the penetrant with the highest $R_d$ value) includes both qualitative aspect and cost. Importantly, the choice of the penetrant is by using two methods of decision support and is also based on compliance assessment with key criteria compatible with sustainability idea and at the same time with Society 5.0 and Industry 4.0 concepts.

The algorithm of the proposed method of choice the penetrant is shown in Figure 1.

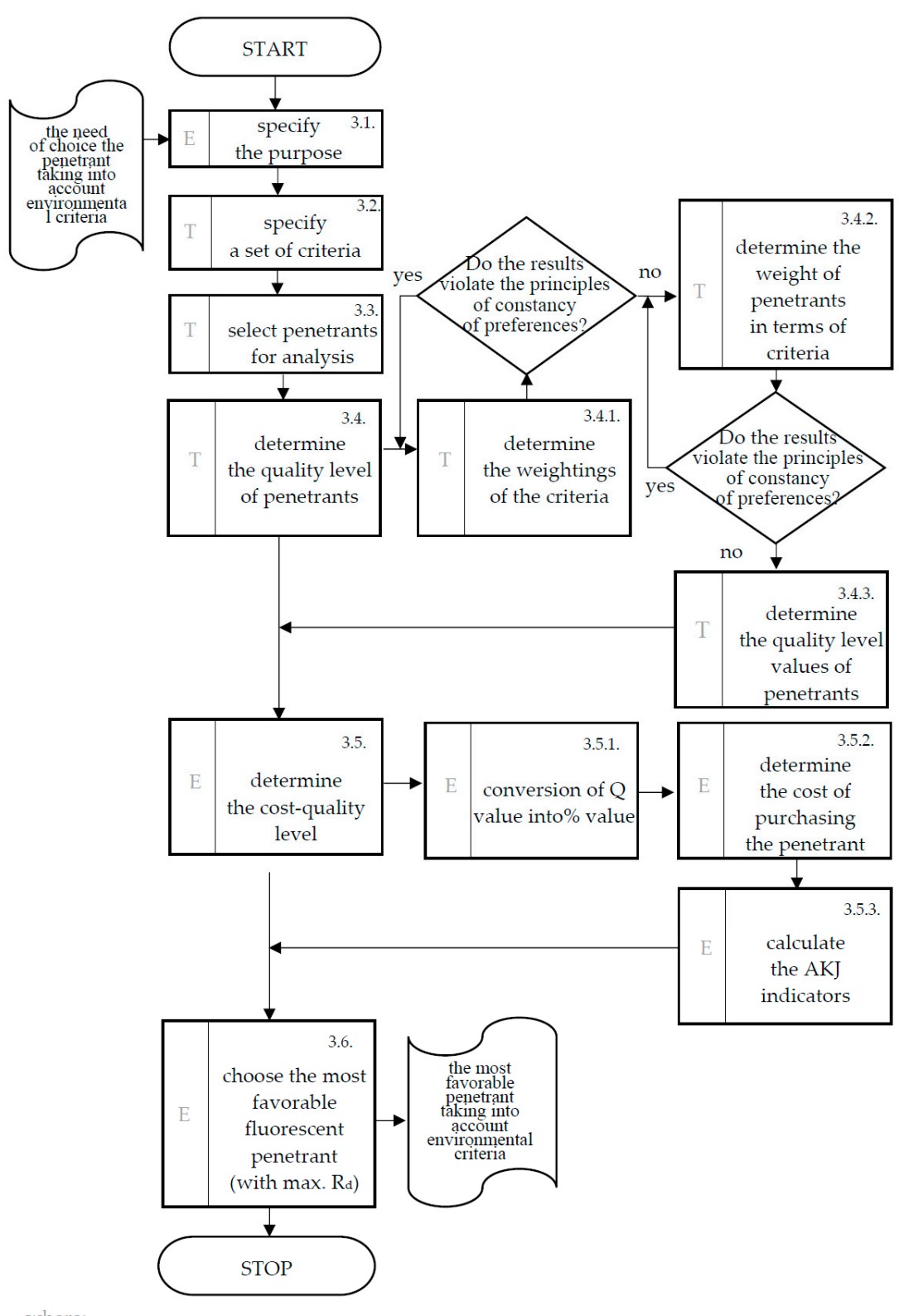

**Figure 1.** An algorithm for choice a penetrant.

In this method, it should be remembered that the types of fluorescent penetrants and ratings awarded to them are dependent on subjective expert judgment; therefore, the results from the method

may be different in terms of preferences of entity using the method, revealing the choice and assessment of products. However, the method itself allows one to choose a fluorescent penetrant.

## 4. Results

The results were analyzed in two steps, i.e., description of the results (Section 4.1) and interpretation of the results and selected conclusions (Section 4.2).

### *4.1. Description of the Results*

With the aim of testing a proposed method to choose the most favorable fluorescent penetrant, which at the same time will ensure the reliability and effectiveness of the FPI research and include the sustainability aspects, the enterprise was selected, which has been struggled with more than once regarding problems of choice the penetrant. Hitherto, the inspector of non-destructive testing made the choice of penetrant; this is not an isolated phenomenon in enterprises and results from the choice of penetrant being made mainly by the criterion of efficiency and price. The inspection was based on their knowledge and experience. During the choice, they were taking into account, among others, internal specifications, economic aspects (the cost of purchasing penetrants), product construction, penetration time and also requirements of external clients ordering product inspections. However, because of constant changes in the environment and in technological processes, it was a must to improve the current choice of penetrants. Therefore, the aim of the test was to propose the method of choice of the fluorescent penetrants for FPI research, using the AHP and AKJ methods and including, through the criteria, the idea of sustainable development, which involves the leading ideas of Society 5.0 and Industry 4.0 concepts.

In the first stage, the aim was specified, which was to support the decision process by the choice of the fluorescent penetrant for FPI research, while including sustainability criteria.

Subsequently, using the brainstorm and multiple voting technique, a set of criteria was created, on the basis of which it was decided to choose a fluorescent penetrant. The team of experts consisted of the manager of the NDT research and the authors of the work, resulting from the fact that very few studies has covered this issue because there is hardly any previous research on the topic and insufficient empirical observations to turn it into a quantitative study. [54] After brainstorm and the multiple voting technique, it was considered that, from point of view of the FPI research conducted in the enterprise, it is necessary to include, in addition to the criteria indicated in the context of sustainable development, i.e., influence on natural environment, reactivity, combustibility, and also additional criteria, i.e.,: level of the sensitivity and type of washing (emulgation). The criterion of the level of penetrant sensitivity was considered as important in the framework of effective FPI research because it regulates the size of the detected defects, which makes it possible to reduce the incorrect rejection of a product or acceptance of an incompatible product, causing failure or its withdrawal [55–57]. In addition, the types of washing (emulgation) which generate effective FPI research (regarding such factors as, for example, costs and research time) which stood out were the washing with water, solvent and emulsifier types [34].

Next, the selected team of experts (the manager of the NDT and the authors of the work), based on knowledge and experience, made the choice of fluorescent penetrants to analyze. It was considered, that the choice of the fluorescent penetrants about the average level of sensitivity (level equal 2) is effective in ongoing the FPI research in the enterprise. According to the selected method, the penetrants which were chosen based on the knowledge and experience were ZL-2C and ZL-60D (Magnaflux), FP-922 (Met-L-Chek), Ardrox 970P23 (Chemetall), FP-22B and MH-3A (Sherwin Incorporated). Penetrants for the needs of the test were randomly and conventionally marked P1–P6.

### *4.2. Interpretation of the Results and Selected Conclusions*

In the subsequent stage of the method, the quality level of fluorescent penetrants by the AHP method was appointed.

In this case, also in view of the fact that very few research has covered this issue until now, to turn it into a quantitative study, the grades of all the criteria in the set of criteria were granted according to Saaty scale [40,41] (Table 1) by the team of experts (i.e., the manager of the NDT and the authors of the work) by the multiple voting technique.

**Table 1.** Awarded grades of all criteria from the set of criteria.

| Criterion | Grade | Description of Grade |
|---|---|---|
| Level of sensitivity | 9 | Absolutely preferred |
| Natural environment | 7 | Very strongly preferred |
| Type of washing (emulgation) | 5 | Strongly preferred |
| Reactivity | 3 | Moderate preferred |
| Combustibility | 3 | Moderate preferred |

Then, based on awarded grades to criteria, the Saaty matrix was made (Table 2).

**Table 2.** The Saaty matrix for each criterion from a set of criteria.

| Criterion | Level of Sensitivity | Natural Environment | Type of Washing (Emulgation) | Reactivity | Combustibility |
|---|---|---|---|---|---|
| Level of sensitivity | 1 | 9/7 | 9/5 | 9/3 | 9/3 |
| Natural environment | 7/9 | 1 | 7/5 | 7/3 | 7/3 |
| Type of washing (emulgation) | 5/9 | 5/7 | 1 | 5/3 | 5/3 |
| Reactivity | 3/9 | 3/7 | 3/5 | 1 | 1 |
| Combustibility | 3/9 | 3/7 | 3/5 | 1 | 1 |

Next, in order to determine the weights of criteria, a calculation was made and it was checked whether the results violate the principle of the constancy of preferences (Table 3). During calculation, the average value of random index, according to Saaty, was equal to r = 1, 12, in view of that number of criteria which were analyzed which equaled five, i.e., n = 5 [50].

**Table 3.** Results of AHP calculation for criteria from a set of criteria.

| Criterion | Sum of Grades | Weight | Coefficient | | |
|---|---|---|---|---|---|
| | | | $\lambda_{max}$ | CI | CR |
| Level of sensitivity | 1.667 | 0.333 | | | |
| Natural environment | 1.296 | 0.259 | | | |
| Type of washing (emulgation) | 0.926 | 0.185 | 5 | 0 | 0 |
| Reactivity | 0.556 | 0.111 | | | |
| Combustibility | 0.556 | 0.111 | | | |

After making calculations and checking them, complete compliance of dominance and consistency assessments were demonstrated, where $\lambda_{max}$ = n and *CI* = 0 and *CR* = 0.

Then, an expert (the manager of the NDT) and the authors of the work made the assessment of penetrants by the criteria according to the Saaty scale, using the multiple voting technique. During the assessment, the average level of sensitivity was omitted because this was the same for all the penetrants (Table 4).



**Table 4.** Assessment of the penetrants by criteria.

| Criterion | Type of Penetrant and Grade | | | | | |
|---|---|---|---|---|---|---|
| | **P1** | **P2** | **P3** | **P4** | **P5** | **P6** |
| Natural environment | 5 | 1 | 3 | 3 | 5 | 5 |
| Type of washing (emulgation) | 5 | 7 | 7 | 5 | 3 | 9 |
| Reactivity | 9 | 7 | 7 | 7 | 9 | 5 |
| Combustibility | 7 | 7 | 3 | 5 | 7 | 3 |

Next, based on the grades awarded to penetrants by the criteria, the Saaty matrix was made (Table 5).

**Table 5.** Saaty matrix for fluorescent penetrants assessment by criteria.

| **Natural Environment** | **P1** | **P2** | **P3** | **P4** | **P5** | **P6** |
|---|---|---|---|---|---|---|
| P1 | 1 | 5 | 5/3 | 5/3 | 1 | 1 |
| P2 | 1/5 | 1 | 1/3 | 1/3 | 1/5 | 1/5 |
| P3 | 3/5 | 3 | 1 | 1 | 3/5 | 3/5 |
| P4 | 3/5 | 3 | 1 | 1 | 3/5 | 3/5 |
| P5 | 1 | 5 | 5/3 | 5/3 | 1 | 1 |
| P6 | 1 | 5 | 5/3 | 5/3 | 1 | 1 |
| **Type of Washing (Emulgation)** | **P1** | **P2** | **P3** | **P4** | **P5** | **P6** |
| P1 | 1 | 5/7 | 5/7 | 1 | 5/3 | 5/9 |
| P2 | 7/5 | 1 | 1 | 7/5 | 7/3 | 7/9 |
| P3 | 7/5 | 1 | 1 | 7/5 | 7/3 | 7/9 |
| P4 | 1 | 5/7 | 5/7 | 1 | 5/3 | 5/9 |
| P5 | 3/5 | 3/7 | 3/7 | 3/5 | 1 | 3/9 |
| P6 | 9/5 | 9/7 | 9/7 | 9/5 | 9/3 | 1 |
| **Reactivity** | **P1** | **P2** | **P3** | **P4** | **P5** | **P6** |
| P1 | 1 | 9/7 | 9/7 | 9/7 | 1 | 9/5 |
| P2 | 7/9 | 1 | 1 | 1 | 7/9 | 7/5 |
| P3 | 7/9 | 1 | 1 | 1 | 7/9 | 7/5 |
| P4 | 7/9 | 1 | 1 | 1 | 7/9 | 7/5 |
| P5 | 1 | 9/7 | 9/7 | 9/7 | 1 | 9/5 |
| P6 | 5/9 | 5/7 | 5/7 | 5/7 | 5/9 | 1 |
| **Combustibility** | **P1** | **P2** | **P3** | **P4** | **P5** | **P6** |
| P1 | 1 | 1 | 7/3 | 7/5 | 1 | 7/3 |
| P2 | 1 | 1 | 7/3 | 7/5 | 1 | 7/3 |
| P3 | 3/7 | 3/7 | 1 | 3/5 | 3/7 | 1 |
| P4 | 5/7 | 5/7 | 5/3 | 1 | 5/7 | 5/3 |
| P5 | 1 | 1 | 7/3 | 7/5 | 1 | 7/3 |
| P6 | 3/7 | 3/7 | 1 | 3/5 | 3/7 | 1 |

Subsequently, based on a matrix of pair comparisons the penetrants (Saaty matrix), the calculations were made, and it was checked whether the results obtained did not violate the principle of the constancy of preferences (Table 6). During the calculation, the average value of random index, according to Saaty, was equal to r = 1, 24, because the number of fluorescent penetrants which were analyzed was six, i.e., n = 6 [50].

**Table 6.** Results of AHP calculation for penetrants.

| Penetrant | Sum of Grades | Weight | Coefficient | | |
|---|---|---|---|---|---|
| | | | $\lambda_{max}$ | CI | CR |
| **Natural environment** | | | | | |
| P1 | 1.364 | 0.227 | | | |
| P2 | 0.273 | 0.045 | | | |
| P3 | 0.818 | 0.136 | 6 | 0 | 0 |
| P4 | 0.818 | 0.136 | | | |
| P5 | 1.364 | 0.227 | | | |
| P6 | 1.364 | 0.227 | | | |
| **Type of washing (emulgation)** | | | | | |
| P1 | 0.833 | 0.139 | | | |
| P2 | 1.167 | 0.194 | | | |
| P3 | 1.167 | 0.194 | 6 | 0 | 0 |
| P4 | 0.833 | 0.139 | | | |
| P5 | 0.500 | 0.083 | | | |
| P6 | 1.500 | 0.250 | | | |
| **Reactivity** | | | | | |
| P1 | 1.227 | 0.205 | | | |
| P2 | 0.955 | 0.159 | | | |
| P3 | 0.955 | 0.159 | 6 | 0 | 0 |
| P4 | 0.955 | 0.159 | | | |
| P5 | 1.227 | 0.205 | | | |
| P6 | 0.682 | 0.114 | | | |
| **Combustibility** | | | | | |
| P1 | 1.313 | 0.219 | | | |
| P2 | 1.313 | 0.219 | | | |
| P3 | 0.563 | 0.094 | 6 | 0 | 0 |
| P4 | 0.938 | 0.156 | | | |
| P5 | 1.313 | 0.219 | | | |
| P6 | 0.563 | 0.094 | | | |

After making these calculations and checking them, the complete compliance of dominance and consistency assessments were demonstrated, where $\lambda_{max}$ = n and *CI* = 0 and *CR* = 0. Then, based on the values of criteria, the ranking of the criteria was made (Table 7).

**Table 7.** Ranking of all criteria from a set of criteria.

| Criterion | Weight | Ranking |
|---|---|---|
| Level of sensitivity | 0.333 | 1 |
| Natural environment | 0.259 | 2 |
| Type of washing (emulgation) | 0.185 | 3 |
| Reactivity | 0.111 | 4 |
| Combustibility | 0.111 | 4 |

The most important criterion was the level of sensitivity with weight 0.33, then the natural environment (0, 259), then the type of washing (emulgation) (0, 185), and next, the reactivity and combustibility with the same weight equal 0.111.

Then, the weight values corresponding to a given criterion were multiplied by all the penetrant weight values relative to a given criterion weight. The values obtained were added up, and the sums obtained were the quality level of penetrants (Q). Next, based on the values of the quality level of the penetrants, the ranking of penetrants was made (Table 8). The criterion of level of sensitivity was omitted because for each of the penetrants, the average level of sensitivity was the same.

**Table 8.** Comparison of criteria weights with penetrant weights in terms of criteria.

| Type of Penetrant | Natural Environment | Type of Washing (Emulgation) | Reactivity | Combustibility | |
|---|---|---|---|---|---|
| | 0.259 | 0.185 | 0.111 | 0.111 | WEIGHT of criteria |
| P1 | 0.227 | 0.139 | 0.205 | 0.219 | |
| P2 | 0.045 | 0.194 | 0.159 | 0.219 | |
| P3 | 0.136 | 0.194 | 0.159 | 0.094 | WEIGHT of penetrants |
| P4 | 0.136 | 0.139 | 0.159 | 0.156 | |
| P5 | 0.227 | 0.083 | 0.205 | 0.219 | |
| P6 | 0.227 | 0.250 | 0.114 | 0.094 | |

| Type of Penetrant | Natural Environment | Type of Washing (Emulgation) | Reactivity | Combustibility | Sum (Q) | Ranking |
|---|---|---|---|---|---|---|
| P1 | 0.059 | 0.026 | 0.023 | 0.024 | 0.132 | 1 |
| P2 | 0.012 | 0.036 | 0.018 | 0.024 | 0.090 | 6 |
| P3 | 0.035 | 0.036 | 0.018 | 0.010 | 0.099 | 4 |
| P4 | 0.035 | 0.026 | 0.018 | 0.017 | 0.096 | 5 |
| P5 | 0.059 | 0.015 | 0.023 | 0.024 | 0.121 | 3 |
| P6 | 0.059 | 0.046 | 0.013 | 0.010 | 0.128 | 2 |

The most favorable fluorescent penetrant in terms of quality, having the maximum value Q, proved to be penetrant conventionally marked P1, of which the sum of values, and so the quality level (Q), was equal 0.132.

In the next stage of the proposed method, aiming to include the cost of the purchase of the penetrants, the cost–quality analysis (AKJ) was made.

At the beginning of the calculations, the values of the quality level of the penetrants were converted into percentages. Then, the cost of purchasing the penetrants was estimated. The cost of the purchase of one package of fluorescent penetrant was adopted. Because of the significant amounts of penetrant consumed in the enterprise (average from 1000 to 14,000 dm$^3$/year/2 penetrants), the cost was calculated for a maximum available capacity of fluorescent penetrants, so 208 dm$^3$.

In the next step, the indicators using in the AKJ method were calculated. Results from the calculation are shown in Table 9.

**Table 9.** Results from a cost-quality analysis.

| Indicator | Type of Penetrant and a Result of Indicator | | | | | |
|---|---|---|---|---|---|---|
| | P1 | P2 | P3 | P4 | P5 | P6 |
| Cost P, €/package | 3094.85 | 2389.8 | 2022.09 | 3026.58 | 4486.34 | 1921.40 |
| Quality Q [%] | 13.168 | 8.978 | 9.946 | 9.611 | 12.139 | 12.826 |
| $c_k$ | 235.036 | 266.199 | 203.317 | 314.903 | 369.588 | 149.803 |
| $k$ | 0.543 | 0.817 | 0.961 | 0.569 | 0 | 1 |
| $E$ | 4.120 | 9.105 | 9.660 | 5.921 | 0.000 | 7.797 |
| $c$ | 0.612 | 0.470 | 0.757 | 0.249 | 0 | 1 |
| $d$ | 0.879 | 0.945 | 0.948 | 0.916 | 0.000 | 0.936 |
| $R_t$ | 0.423 | 0.417 | 0.471 | 0.367 | 0.065 | 0.518 |
| $R_e$ | 0.579 | 0.694 | 0.848 | 0.499 | 0.008 | 0.934 |
| $R_d$ | 0.501 | 0.556 | 0.659 | 0.433 | 0.036 | 0.726 |
| Place according to $R_d$ | 4 | 3 | 2 | 5 | 6 | 1 |
| Interpretation of the stan | moderate | satisfactory | beneficial | acceptable | definitely unfavorable | definitely the most beneficial |

Based on the results, the fluorescent penetrant was chosen which was most favorable for the enterprise where the method was tested. The choice has been made based on the highest value of the decision-making indicator values ($R_d$). Despite that the difference between penetrants P6 and P3 was only 0.067, according to the idea of the AKJ method, it was preferred to choose the maximum value. Therefore, as it has been shown, the most favorable penetrant in terms of cost–quality proved to be the penetrant conventionally marked P6 ($R_d$ = 0.726).

## 5. Discussion

Changes occurring in the environment, but also in technological processes, forced us to look for contemporary methods supporting the achievement of reliable FPI research, the methods of which must be both effective and consistent with the idea of sustainable development [23–26,57–61]. This search for an adequate method refers to a search for effective, concentrated and divergent solutions of problems where there are complex societal and engineering factors. It also refers to changes taking place in the surroundings, as well as in technological processes [15,16,27,32,57,59,62,63], which enforce the need to search for modern methods that support the achievement of reliable FPI research. After reviewing the literature, it was concluded that the actions taken were to achieve reliable and effective fluorescent penetration inspection [10,20,21], and thus referred to the idea of sustainable development, the concept of Society 5.0 and partly the concept of Industry 4.0. These actions were mainly aiming to reduce the negative influences that fluorescent penetrants have on the natural environment [13,14,22] and reduce the human factor in the decision-making process [10,20,21]. The solution which achieved, to allow one to unambiguously to choose the FPI method, takes into account reliability, efficiency and the environment. In turn, changes in society and industry mean that there is still a need to look for methods of choosing a penetrant. In addition, due to the significant amounts of penetrants consumed in industry, and often automated FPI research [1,3,10], there is a need to made and use the method support the choice of fluorescent penetrant. Among others, the important part of the development of enterprises towards Industry 4.0 is that it allows the choice of a penetrant as a part of digital networks and also with using the programs supporting the decision-making process [15,31]. Therefore, it was concluded that the actions, in terms of the search for a method of choice the penetrants to FPI research, are still a current research problem.

Therefore, the aim of the work was to propose the method of support choice of the fluorescent penetrant to FPI research. In the results of the usage of the proposed procedure, it will be possible to reduce the negative impact of the environment in FPI processes (sustainability), while including other criteria, i.e., financial, security, productive (Industry 4.0) and society (Society 5.0). The essence of the proposed method was to integrate two methods of decision-support, i.e., the AHP method (analytic hierarchy process) and the AKJ (cost–quality analysis), and including in the method the sustainable development criteria. The method was supported by additional techniques, i.e., brainstorming (BM) and multiple voting. In the analysis, the additional levels of the sensitivity criterion and the fluorescent penetrants were included, with an average level of sensitivity (equal 2) were analyzed, i.e., ZL-2C and ZL-60D (Magnaflux), FP-922 (Met-L-Chek), Ardrox 970P23 (Chemetall), FP-22B and MH-3A (Sherwin Incorporated), which were marked randomly and conventionally from P1 to P6. By the AHP method, the level of quality of the fluorescent penetrants was calculated, and by the AKJ method it was calculated, and in the same way identified, the most benefiting penetrant in terms of cost–quality.

Integration of the AHP and AKJ methods resulted from their effectiveness in integrating with other techniques [35–37,44] in the aim of being effective for making the best choice, and also to calculate the level of product quality [38], with included the cost aspect [38,44]. It was concluded that the AHP method is effective because it allows one to analyze any number of products and that the criteria of choice can be quantitative or qualitative, and also these can be analyzed in a subjective and objective way [39–41]. Furthermore, in the context of this method, it was considered important that this AHP method allows the modeling of the decision in the context their uncertainty and risk [36,42] and allows implementation into digital systems, including, for example, the Expert Choice software [31], which refers to enterprise actions as part of Industry 4.0 concept [15].

The main benefit of the proposed method, consisting of integrated the AHP method and AKJ method, is as far as possible subjective support of the choice of the least harmful fluorescent penetrant for the natural environment. The choice of penetrant would not only take place in terms of environmental protection, but would also take into account other criteria, e.g., financial or situation-specific criteria. This choice of criteria in the proposed method refers to criteria of the NFPA and HMIS systems [18,46,47] coexisting as part of sustainability and answering the goals of sustainability, for example, goals 12,

13 and 14 [10–14,27]. This fact is extremely important due to the significant amounts of penetrants used in commonly used FPI research [1,3,10]. In this context, the proposed method refers to the Society 5.0 concept, in which the task is the assurance of good societal well-being while respecting ecological integrity [32]. An important advantage of the proposed method (which is a combination of AHP with AKJ) is the possibility of applying it within digital network systems, giving the opportunity for the development of enterprises towards Industry 4.0 [33]. Establishing the procedures can be conducive to the preparation of the enterprise to the conditions of Industry 4.0. Production engineering deals with such problems. In the context of future works, it should be planned to improve the approaches in assessing criteria and improve the method itself, for example, as part of digital networks using, for example, the Expert Choice program. Next, it is important that future methods would make the comparative analysis of effectiveness in the software based on the proposed method, with other methods of choice of fluorescent penetrants that are currently used in these types of enterprises (making the FPI).

The results obtained indicate that the proposed method can be used to choose fluorescent penetrants in productions and services enterprises which realize fluorescent penetrant inspection, and also in the enterprises that want to develop as part of Industry 4.0.

## 6. Conclusions

Performing non-destructive testing (NDT) and, in particular, fluorescent penetrant inspection (FPI) is still a challenge; therefore, the aim of this work was to propose the method of supporting the choice of a fluorescent penetrant for FPI research. According to the results of the usage of the proposed procedure, it is possible to reduce the negative impacts of the environment of FPI processes (sustainability) while including other criteria, i.e., financial, security, productive (Industry 4.0) and society (Society 5.0). This method was proposed in an enterprise located in Poland, in Podkarpacie, which conducted fluorescent penetration inspection for the aviation industry products. As a part of the proposed method, the experts of the appointed team, in addition to those previously used by the entity, also took into account pro-environmental criteria. The full criteria were: influence on the natural environment, reactivity, combustibility and cost of purchase the penetrant, which were selected in the context of sustainability. The method consisted of the successive methods of AHP and AKJ, and was based on the experts' teamwork, which chose the penetrant assessment criteria and their weight and also made the comparison in the pairs. This is important as the criteria not only used cost and performance criteria, but also environmental criteria (Society 5.0) or criteria to facilitate the introduction to Industry 4.0. Based on calculations made, according to the proposed method, it turned out that the most favorable fluorescent penetrant, in terms of quality, was the penetrant conventionally marked P1. Its level of quality was equal to $Q = 0.132$. In turn, after including the cost of the purchase of the penetrants in AKJ, it was shown that the most favorable penetrant, in terms of cost–quality proved to be penetrant conventionally marked P6 ($R_d = 0.726$). The proposed method can be used to support the decision-making regarding choosing a penetrant for FPI research. It is possible to include the environmental criteria, based on the idea of Society 5.0. The algorithm allows, in future, automation in decision-making which would be necessary from as part of the pursuit of the concept of Industry 4.0. In the use of this method in the particular enterprise studied in this paper, it was shown that the best penetrant, in qualitative terms, was the penetrant marked as P1 (its level of quality was equal $Q = 0.132$). It has not been the most frequently used penetrant so far. Environmental criteria have changed the bases on which decisions are made. After the application of the AKJ method, i.e., the inclusion of the cost of the purchase the penetrants, it was shown that the best penetrant is a different penetrant, marked P6 ($R_d = 0.726$).

Thus, the application of the method may influence the choices of penetrants made, if one takes into account, for example, environmental criteria (Society 5.0) or other criteria, e.g., those preparing for Industry 4.0. In this method, it should be remembered that the types of fluorescent penetrants and ratings awarded to them depend on subjective expert judgment, which, in the future, can be

standardized by the use of, for example, study-specific selection tables. The proposed method can be also used to support decision-making in enterprises.

**Author Contributions:** Conceptualization, A.P. and D.S.; methodology, A.P. and D.S.; validation, D.S., and L.B.; formal analysis, D.S. and A.P; investigation, D.S.; resources, L.B.; data curation, D.S.; writing—original draft preparation, D.S. and A.P.; writing—review and editing, L.B. and A.P.; visualization, D.S.; supervision, A.P.; funding acquisition, L.B. All authors have read and agreed to the published version of the manuscript.

**Funding:** This research was funded by Faculty of Mining, Ecology, Process Control and Geotechnologies (Faculty BERG), Technical Unviersity of Košice, Slovakia.

**Acknowledgments:** We wish to appreciate the reviewers for the valuable suggestions that helped to significantly improve the quality of our revised manuscript.

**Conflicts of Interest:** The authors declare no conflict of interest.

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
