# Peer review of "Method of Choice: A Fluorescent Penetrant Taking into Account Sustainability Criteria"

_sustainability, doi:10.3390/su12145854_

Round 1

Reviewer 1 Report

happy to accept it.

Reviewer 2 Report

This can be accepted as a case report  but not as a research paper.

In this context, the improvement is notorious and the text can be accepted

Reviewer 3 Report

Dear Authors 

Thank you for responding to the comments of the reviewers.

All the best 

This manuscript is a resubmission of an earlier submission. The following is a list of the peer review reports and author responses from that submission.

Round 1

Reviewer 1 Report

GENERAL COMMNETS:

The authors propose an interesting method to evaluate products through AHP and AKJ. 

But, the benefits of this method on sustainability, society 5.0 or industry 4.0 are not justified. That is, it could be applied to these objectives or others. It seems that the concept Industry 4.0 and society 5.0 are introduced because these are now "trending". But: How do the results of this paper contribute to the objectives of sustainability, society 5.0 or industry 4.0? This is not justified.

SPECIFIC COMMENTS

INTRODUCTION: It would be interesting to indicate in the introduction why these two integrated methods contribute to Industry 4.0

METHODS: A schematic or flow chart would be interesting to better understand the procedure.

Section 3.2. Sustainable criteria should be explained; the choice should also be justified according to the references given by the authors [6, 9, 10, 11] and why. How is the effectiveness of improving these sustainable criteria known? Data on environmental impact? which?

Section 3.4.2. How is  this carried out? :"An assessment and weights should be carried out for each selected fluorescent penetrant";  With expert judgment? indicate and clarify

Final comments:

One of the problems in evaluating the sustainability of products is identifying the evaluation criteria: medium impacts, damages generated, value created?

Reviewer 2 Report

Dear authors,

The paper needs an extensive English review. Please make the following changes.

Line 18: with included the 19 sustainability should be which included the 19 sustainability

Line 41: Add , after sustainability

Line 43: Rewrite the sentence

Line 72, 73, and 74: Rewrite the sentence

Line 76: However, it was observed,. Comma is not required after observed.

Line 259: He was based or his method was based?

You can discuss what other decision methods have been used other than AHP and AKJ.

Reviewer 3 Report

In view of this manuscript, I have the following comments:   1. hightlighs are not addressed properly.    2.the computation and experiment facility should be given with more details for the numerical simulation and validation.    3. Conclusion and directs for future researches section, has not been properly organised.   4. In the future work, could a general computational intelligence aided design framework be utilised in the smart design process?

Reviewer 4 Report

Title:
OK.

Abstract:
The structure followed by authors is:
Problem to be solved + Objective + Method + Explanation of Method + Applicability
I recommend to follow the structure proposed by MDPI:
Background + Methods + Results + Conclusions
First, place the question addressed in a broad context and highlight the purpose of the study (at least, define the challenge and connect it with S5 and I4). Then, describe briefly the main methods or treatments applied. Next, summarize the article's main findings (Where they are?). Later, indicate the main conclusions and your interpretations.
I recommend to completely rewrite it in order to focus the problem to be solved, how it is going to be solved and which benefits are got doing it.

Keywords:
Nothing about sustainability, Society 5.0, Industry 4.0 and mechanical engineering. Only the first three are mentioned, but not introduced, explained or developed.

Introduction:
The context has not been defined. There are a lot of NDT, such as industrial radiography tests (to study the internal discontinuities of a material, by means of ionising electromagnetic radiation), penetrant liquid tests (to identify irregularities on the surface of materials that do not have porosity), ultrasound tests (to identify irregularities through the use of acoustic waves) or magnetic particle tests (to observe discontinuities in ferromagnetic materials. A metal powder is subjected to the action of a magnetic field. Discontinuities are observed when metal powder accumulates in a certain area).
Each test has its own purpose. So you should introduce the issue properly. Industrial enviroments, working with metals, etc.
I recommend again to follow the structure proposed by MDPI:
The introduction should briefly place the study in a broad context and highlight why it is important and define the purpose of the work and its significance. Therefore, the current state of the research field should be reviewed carefully and key publications cited.
From line 37 onwards, the paper becomes better.
However, the Society 5.0 and Industry 4.0 concepts introduce noise to the research. (The title sounds better: Method of choice a fluorescent penetrant taking into account sustainability criteria)
Last paragraph has to be improved. ("in context to sustainable development being the leading idea for Society 5.0 and Industry 4.0 conceptions" ?). Do you want to select a fluorescent penetrant according to sustainable development principles? Try to explain it easier.

Materials:
Why are ZL-2C and ZL-60D, FP-922, Ardrox 970P23 114, FP-22B and MH-3A selected to be studied and compared? Only one expert? Is it contrasted in literature? A group the experts should be used. The number of experts should be justified. And the criteria to be considered an expert should be justified too.

Methods:
Integration of AHP and AKJ methods should be explained, justified and referenced (if there is a previous reference).
The choice of AHP method should be highlighted. Idem with AKJ.
In the AHP method is essential the selection of criteria, what is previous to the own method. Other methods as SWOT analysis (or others) can be used... or even a literature review. But this review must be robust. To include [6, 9, 10, 11] is no enough.
Many times the group of experts who are going to solve the AHP can be previously asked about this.
The objective should be SMART defined. (the aim exposed should be refined).
Who is determining the weight of each criterion?
The construction of the ranking is not clear.
This must be better explained:
??=0,0667(8?+4?+2?+?) (11)
??=0,0667(8?+4?+2?+?) (12)
??=(??+??)/2 (13)

Why is 8?+4?+2?+? the ratio to define the technical preference? Why this exponential growth?
Why is 8?+4?+2?+? the ratio to define the economical preference? Why this exponential growth?

At the end, Rd= (9k+9q+6d+6c)/30, Why?

AHP is used by experts. The ratio 1/9 - 9, to 1-1 are not law. They are a reference, a basis which can be modified by experts.
AHP is a multi-criteria decision tool. AHP is applied through the construction of hierarchical structures, in which the first hierarchical level consists of the problem goal, the next hierarchical levels encompass the decision criteria and sub-criteria and, finally, the last level contains the alternatives. For allocating weights to the decision criteria and sub-criteria, paired comparisons are made (criterion vs. criterion) by knowledgeable members of a panel. These numerical comparisons are generally expressed in a 1-to-9 scale and lead to dominance matrices.
Many cases a specific questionnaire is designed to capture the judgements of all experts.
The used given in this research should be stressed.
Who is deciding in the Figure 1?

Results:
To justify "the expert (head of NDT) based on his knowledge and experience, awarded grades to all criteria from a set of criteria according to Saaty scale" is not enough.
All the research is based on the judgement of one persone. Even if various experts from NDT would be selected to decide (getting a consensus), the methodology used would be a Case study. The case of NDT.
The researches which are based based on case studies must be justified. (for example, very few studies have covered this issue because there is hardly any previous research on the topic and insufficient empirical observations to turn it into a quantitative study).
What is the reliability and goodness of P6 against P3. Is there a significant difference?

Discussion:
Conclusions are based on the results from the choice of one person. Only one for one company.
According to MDPI recommendations:
Authors should discuss the results and how they can be interpreted in perspective of previous studies and of the working hypotheses.

Reviewer 5 Report

Dear Authors,

You have done a good job. However, the English language and the presentation style of the paper is very poor. 

I suggest sending the paper to a professional/academic proofreader in order to enhance the language and presentation level.